# Spherization Layer:
# Representation Using Only Angles

**Hoyong Kim, Kangil Kim**[*]
Artificial Intelligence Graduate School
Gwangju Institute of Science and Technology,
Gwangju 61005, South Korea
`hoyong.kim.21@gm.gist.ac.kr, kangil.kim.01@gmail.com`

## Abstract

In neural network literature, angular similarity between feature vectors is frequently used for interpreting or re-using learned representations. However, the inner product in neural networks partially disperses information over the scales and angles of the involved input vectors and weight vectors. Therefore, when using only angular similarity on representations trained with the inner product, information loss occurs in downstream methods, which limits their performance. In this paper, we proposed the *spherization layer* to represent all information on angular similarity. The layer 1) maps the pre-activations of input vectors into the specific range of angles, 2) converts the angular coordinates of the vectors to Cartesian coordinates with an additional dimension, and 3) trains decision boundaries from hyperplanes, without bias parameters, passing through the origin. This approach guarantees that representation learning always occurs on the hyperspherical surface without the loss of any information unlike other projection-based methods. Furthermore, this method can be applied to any network by replacing an existing layer. We validate the functional correctness of the proposed method in a toy task, retention ability in well-known image classification tasks, and effectiveness in word analogy test and few-shot learning. Code is publicly available at `https://github.com/GIST-IRR/spherization_layer`

## 1 Introduction

The inner product is a key element constituting layers in deep neural networks with a nonlinear activation function. The inner product with the Euclidean norms and the angle, that is, $\|\mathbf{w}_i\|\|\mathbf{x}_j\| \cos \theta_{ij}$, has been analyzed in terms of the norms $\|\mathbf{w}_i\|\|\mathbf{x}_j\|$ and the angle $\cos \theta_{ij}$, independently [11, 12, 31]. All factors of the inner product learn distinct information. Therefore, using only one factor results in information loss when re-using or understanding the information in downstream tasks. This problem is termed as *dispersion problem*. The angular similarity between features is frequently used. However, this technique causes the dispersion problem in advanced methods in neural network literature such as decoupled network [12], representation learning [5, 6, 20, 21, 28], regularization [29, 30], zero-shot learning [22], and generative model [3, 23]. To mitigate the dispersion problem, numerous angle-based learning approaches [1, 9, 10, 13, 15, 14, 29, 30, 32] have been proposed. However, these studies are based on projection onto the hyperspherical surface. In this projection method, distinction by the scale of features is ignored. Therefore, information loss occurs.

We proposed the *spherization layer* as an explicit solution for the dispersion to completely eliminate the interference of the norms in training without drawbacks. This layer is used to locate all represen-

---

[*]corresponding author

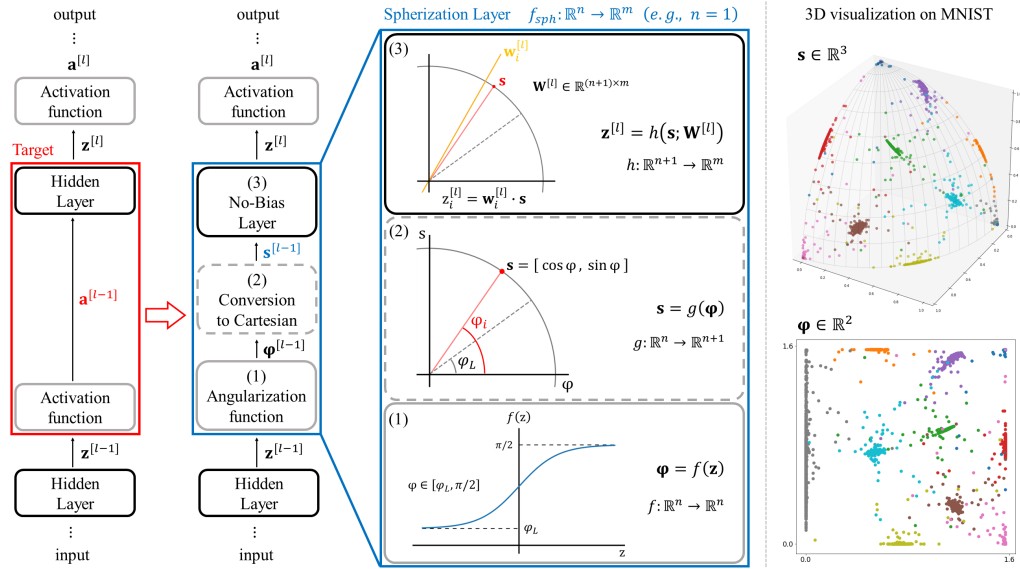

Figure 1: Overview of *spherization layer*. The red box indicates the existing fully connected layer to be replaced with the spherization layer in the blue box. After replacement, the pre-activations from $(l-1)$th layer are passed to the angularization function, not activation function. These pre-activations are converted to angular coordinates in (1). Through (2), spherized representations are located on the $(n+1)$-spherical surface. Finally, a hidden layer without bias parameters is trained on these spherized representations for using only the angles in (3). These relations of the input and output of the stage (1), (2), and (3) are illustrated as graphs, which are clearly defined in Eq. 3, Eq. 6, and Eq. 9, respectively. The right side displays how to generate angular coordinates and convert them to Cartesian coordinates in a 3-dimensional space on MNIST

tations onto a constrained region on the hyperspherical surface and train hyperplanes passing through the origin to learn representations with only the angles.

The spherization layer consists of three main components: *Angularization function* that converts the pre-activations from the previous hidden layer to angles; *Conversion* from spherical coordinates to Cartesian coordinates located on the hyperspherical surface; *No-bias layer*, a hidden layer without bias parameters, that determines decision boundaries by using only the angles. Figure 1 illustrates the design of the spherization layer. Through *spherization*, representations are located on the hyperspherical surface and the effect of the norms in representation learning is completely eliminated. Thus, neural networks are enforced to express all representations differently by using only the angles.

We experimentally verified the functional correctness of the spherization layer in a toy task and its applicability to feedforward and convolutional neural networks by evaluating performance on image classification tasks. The results reveal that the training ability of original networks is preserved after applying the spherization layer. Furthermore, we analyzed the sensitivity to width and depth, the effect of projection in the spherization layer, and the influence of the spherization layer on the gradient flows in training. Finally, we investigated the effect of the proposed method in downstream tasks through visualization, word analogy test, and few-shot learning.

In summary, our contributions are three-fold:

- To address the dispersion problem, we propose the *spherization layer* to represent all feature vectors on the hyperspherical surface and learn the representations with only the angles.

- We validate the wide-applicability and scalability of the spherization layer without any loss of performance through experiments on various well-known networks.

- We empirically show that the spherization layer can be used in many applications in which angular similarity is a critical metric.

## 2 Background

**Conversion Spherical to the Cartesian Coordinate System**  In most neural networks, all input samples on the $n$-dimensional space are represented as Cartesian coordinates, in which $i$-th column value denotes the distance from the origin along the $i$-th axis, and neural networks train them by using neurons in which the inner product between the input and weight vector occurred. The pre-activations from neurons are used to determine whether neurons should be activated by using the following activation function. In this process, the pre-activations are calculated by using weight and bias parameters, represented as Cartesian coordinates. These Cartesian coordinates can be converted from spherical coordinates. Given a vector **s** represented as Cartesian coordinates on the $n$-dimensional space, **s** can be defined as spherical coordinates $\boldsymbol{\theta} = [r, \boldsymbol{\varphi}]$, composed of a radial coordinate $r$ and $n$-1 angular coordinates $\boldsymbol{\varphi} = [\varphi_1, \varphi_2, ..., \varphi_{n-1}]$. In this case, the $k$-th axis of **s** can be computed from $\boldsymbol{\theta}$ with Eq. 1

$$\mathbf{s} \;=\; [r\cos\varphi_1, \cdots, r\cos\varphi_k \textstyle\prod_{i=1}^{k-1}\sin\varphi_i, \cdots, r\prod_{i=1}^{n-1}\sin\varphi_i] \tag{1}$$

, where $s_{1<k<n} = r\cos\varphi_k \prod_{i=1}^{k-1}\sin\varphi_i$.

Generally, the spherical coordinate system is a 3-dimensional version of the polar coordinate system. However, the spherical coordinate system in the followings indicates all $n$-dimensional versions of the polar coordinate system, where $n$ is greater than or equal to 2.

## 3 Spherization Layer

We proposed the *spherization layer* as shown in Figure 1 to locate feature vectors on the hyperspherical surface without any loss of information and learn hyperplanes by using only the angles. In this method, a layer of an original network is selected to learn representations by angular similarity. Next, the layer is replaced to a *spherization layer* through three sequential stages: *Angularization*, *Conversion to Cartesian*, and *No-bias training* notated as $f$, $g$, and $h$ functions, respectively. The final form of spherization layer $f_{sph}$ is expressed as Eq. 2.

$$f_{sph} \;=\; (h \circ g \circ f), \quad f_{sph} : \mathbb{R}^n \to \mathbb{R}^m \tag{2}$$

After training with the layer, we can obtain the representations on the $(n + 1)$-spherical surface, namely spherized representations, as the outputs of $(g \circ f)$ operations.

The common goal of all stages is to preserve the training ability of the original network while the spherization layer trains all information on the $(n + 1)$-spherical surface. The angularization locates all pre-activations on the safe spherical surface. The conversion to Cartesian results in the generation of compatible representations to the ordinary layer. No-bias training enforces training by only the angles. In the followings, we elaborate each stages more detail and only annotate the $(l - 2)$th and $(l)$th layer as $[l - 2]$ and $[l]$, respectively, for the simplicity.

### 3.1 Angularization

$$\boldsymbol{\varphi} \;=\; f(\mathbf{z}), \quad f : \mathbb{R}^n \to \mathbb{R}^n \tag{3}$$

*Angularization* is the stage to map a pre-activation vector $\mathbf{z}$ , passed from $(l - 1)$th layer, to angular coordinates $\boldsymbol{\varphi}$. The $n$ indicates the dimension of the pre-activation vector. The role of this stage is to configure the shape of the mapped region on the $(n + 1)$-spherical surface for resolving training and computational difficulty.

Angularization $f$ is implemented by applying the following element-wise function to all coordinates of $\mathbf{z}$ as an activation function, and $f$ is illustrated as Eq. 4.

$$f(\mathbf{z}) \;=\; \left(\tfrac{\pi}{2} - \varphi_L\right) \cdot \sigma(\alpha \cdot \mathbf{z}) + \varphi_L \tag{4}$$

, where the terms and form are used by three following motivations.

**Converting Pre-Activation to the Angular Coordinate**  The first step is to convert the input vector into angular coordinates. To ensure the conversion as bijective mapping, we restrict the range of the function as $[0, \frac{\pi}{2}]$. To allow unrestricted input representations on the real-valued domain, the sigmoid function $\sigma(\cdot)$ is used with weight $\frac{\pi}{2}$ for the range setting. As the sigmoid function used,

the input vector should be pre-activations, not activations because when activations from ReLU or another sigmoid are passed to the angularization, inefficient use of the spherical surface or gradient amplification, respectively, may occur. After converting pre-activations to angular coordinates, the representations on the hyperspherical surface in the same range were located by setting a consistent radius over all inputs. This radius scale is controlled in the conversion-to-Cartesian stage.

**Tailoring Angular Representation Space**   In the conversion from angular to Cartesian coordinates, the last coordinate can be an extremely small value because trigonometric values in [0, 1] are multiplied many times. This scale descent can map all values in the axis to only a single value by the limit of the floating point data type. To reduce this effect in the angularization, we introduce a lower bound $\varphi_L$ of angles to guarantee distinguishable values in its corresponding converted Cartesian coordinates as the following equation (Eq. 5):

$$\varphi_L = \sin^{-1}\left(\delta^{1/n}\right) \tag{5}$$

, where $\delta$ is a minimal trigonometric value to guarantee the distinguishable representations. The details are presented in Appendix A. We set $\delta$ to the empirically obtained proper value $10^{-6}$, for all experiments.

**Scaling Pre-Activations**   In angularization, the activations are concentrated onto the small region because of the lower bound. This concentration renders training difficult with the decrease in the variance. To reduce the effect, we set a learnable parameter $\alpha$ as a weight of $\mathbf{z}$, which controls the variance of $\mathbf{z}$. Using this scale factor, the generated angular representations become abundant.

### 3.2   Conversion-to-Cartesian

$$\mathbf{s} = g(\boldsymbol{\varphi}), \quad g : \mathbb{R}^n \to \mathbb{R}^{n+1} \tag{6}$$

In the *Conversion-to-Cartesian* stage, the angular coordinates of the previous stage are converted to Cartesian coordinates on the $(n + 1)$-spherical surface. This conversion ensures the consistency between the output of angularization (polar coordinate system) and the input of the following no-bias layer (Cartesian coordinate system), and enables the layer to be trained in the same way as general neural networks. Furthermore, an additional dimension makes the spherization layer have enough capacity to be compatible with the ordinary layer. This implementation is based on Eq. 1 with the modified range of angles as the following equation (Eq. 7).

$$g(\boldsymbol{\varphi}) = [r\cos\varphi_1, \cdots, r\cos\varphi_k \textstyle\prod_{i=1}^{k-1}\sin\varphi_i, \cdots, r\prod_{i=1}^{n}\sin\varphi_i], \quad \varphi_i \in \left[\varphi_L, \tfrac{\pi}{2}\right] \tag{7}$$

**Calculation Trick**   Implementation of Eq. 7 as a tensor operation requires the trick defined in the following equation (Eq. 8):

$$\begin{aligned}\boldsymbol{\phi} &= \mathbf{W}_\varphi^\top \boldsymbol{\varphi} \\ \mathbf{s} &= r \cdot \exp\left(\mathbf{W}_\phi^\top \ln\left(\sin\boldsymbol{\phi}\right) + \ln\left(\cos\left(\boldsymbol{\phi} + \mathbf{b}_\phi\right)\right)\right)\end{aligned} \tag{8}$$

, where $\boldsymbol{\phi}$ is a dimension-expanded vector in $\mathbb{R}^{n+1}$ and $\phi_{n+1} = \phi_n$, and $r$ is a constant to control radius. Here, $\mathbf{W}_\varphi, \mathbf{W}_\phi$, and $\mathbf{b}_\phi$ are constant matrices and vector in $\mathbb{R}^{n\times(n+1)}, \mathbb{R}^{(n+1)\times(n+1)}$, and $\mathbb{R}^{(n+1)}$, respectively. $\mathbf{W}_\varphi = [\mathbf{I}_n; \mathbf{v}]$, where $\mathbf{v} = [\mathbf{0}; 1]^\top \in \mathbb{R}^n$, is used for matching the dimension between $\boldsymbol{\varphi}$ and $\mathbf{s}$. The other constants are used for calculating logarithm trigonometric values from expanded angular coordinates in the way of matrix multiplication, where $\mathbf{W}_\phi$ is an upper triangular matrix in which all diagonals are zero and $(\mathbf{W}_\phi)_{n,n+1}$ is also zero, and $\mathbf{b}_\phi = [\mathbf{0}; -\tfrac{\pi}{2}]^\top$. See Appendix B for detail hyperparameters and process.

### 3.3   No-bias Training

$$\mathbf{z}^{[l]} = h(\mathbf{s}), \quad h : \mathbb{R}^{n+1} \to \mathbb{R}^m \tag{9}$$

Through angularization and conversion-to-Cartesian, all representations are located on the $(n + 1)$-spherical surface. In this case, any update to the representations is determined by the change of angular similarity. However, hyperplanes on the $(n + 1)$-dimensional space from the next layer may not use only the angular similarity, which may assign semantic information to the Euclidean norm of the spherized representation. To synchronize these parameters, we used *no-bias training* using only the weight parameters $\mathbf{W}^{[l]}$ as illustrated in Eq. 10.

$$\mathbf{z}^{[l]} = \mathbf{W}^{[l]\top}\mathbf{s} \tag{10}$$

**Effect of No-Bias on Training** In the ordinary layer, the problem of no-bias is that hyperplanes passing through the origin cannot be shifted to another parallel hyperplanes. However, the problem disappears when all feature vectors are located on the $(n + 1)$-spherical surface because the decision boundary can be shifted by only the angle changes of $(n + 1)$-dimensional hyperplanes even though they pass through the origin.

### 3.4 Optimization with Overall Process

A training loss $L$ is calculated in the same manner as the original network. The gradient of $L$ for the spherization layer is calculated by multiplying the following partial derivative $\frac{\partial L}{\partial \mathbf{W}}$ to the original backpropagation step from the $(l-2)$th to $(l)$th layer, as illustrated in Eq. 11. The first term $\frac{\partial L}{\partial \mathbf{z}^{[l]}}$ is calculated by the subsequent layers in the same way of the original network.

$$
\begin{aligned}
\frac{\partial L}{\partial \mathbf{W}} &= \frac{\partial L}{\partial \mathbf{z}^{[l]}} \cdot \frac{\partial \mathbf{z}^{[l]}}{\partial \mathbf{s}} \cdot \frac{\partial \mathbf{s}}{\partial \boldsymbol{\varphi}} \cdot \frac{\partial \boldsymbol{\varphi}}{\partial \mathbf{z}} \cdot \frac{\partial \mathbf{z}}{\partial \mathbf{W}} \\
\frac{\partial \mathbf{z}}{\partial \mathbf{W}} &= \mathbf{a}^{[l-2]} \\
\frac{\partial \boldsymbol{\varphi}}{\partial \mathbf{z}} &= \left(\frac{\pi}{2} - \varphi_L\right) \cdot \alpha \cdot \sigma'(\alpha \cdot \mathbf{z}) \\
\frac{\partial \mathbf{s}}{\partial \boldsymbol{\varphi}} &= [-r \sin \varphi_1, \cdots, -r \prod_{i=1}^{k} \sin \varphi_i, \cdots, r \cos \varphi_n \prod_{i=1}^{n-1} \sin \varphi_i] \\
\frac{\partial \mathbf{z}^{[l]}}{\partial \mathbf{s}} &= \mathbf{W}^{[l]}
\end{aligned}
\tag{11}
$$

## 4 Experiments

First, we define two terms for simplicity on indicating two networks: one is the network before applying the spherization layer, called *original network*, and the other is the network after substituting an ordinary layer with the spherization layer, called *spherized network*. In the followings, we used these two terms consistently. All experiments were performed five times with random seeds and their training and test accuracy were evaluated except word analogy test and few-shot learning. The mean $\mu$ and standard deviation $\sigma$ of accuracy are represented as $\mu \pm \sigma$ in each table.

### 4.1 Functional Correctness Test on a Toy Task

**Implementation Details** We verified the spherization layer for learning decision boundaries on a simple binary classification task. We set up the simple binary classification task: given $(\mathbf{x}_i, y_i)$ pairs, where $\mathbf{x}_i$ is the $i$-th input sample in $\mathbb{R}^2$ and $y_i \in \{0, 1\}$ is the label of $\mathbf{x}_i$, we randomly generated 100 input samples located around $(0, 0)$ for the label 0, and the other 100 samples for the label 1 around $(1, 1)$, as shown in Figure 2. We set a 2-layer neural network as the original network, and trained it with the softmax function, cross-entropy, and SGD at a learning rate of 0.01. We applied the proposed method to the original network by replacing the

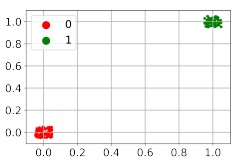

Figure 2: Input samples for the toy task

last fully connected layer with the spherization layer. For comparison of representations in the same dimensional space, we set the dimension of the spherization layer to 1, where it is 2 in the original network. The other settings are identical to the original network. We trained both networks on the input samples for 100 epochs with 16 mini-batches, where the networks converged and achieved 100% training accuracy.

**Distribution of Representations** Initially, all representations are randomly distributed into two groups on a 2-dimensional space. After convergence, the feature vectors are divided into two disjointed groups, as illustrated in Figure 3. This means the spherized network locates all representations on the 2-spherical surface in both the initial and final epoch, whereas the original network spreads them out.

---

[2]https://github.com/weiaicunzai/pytorch-cifar100

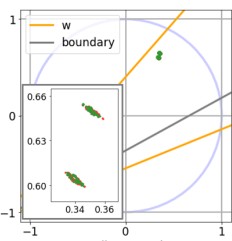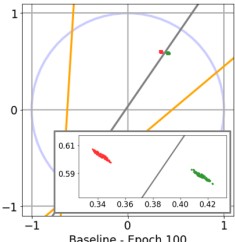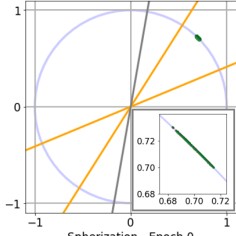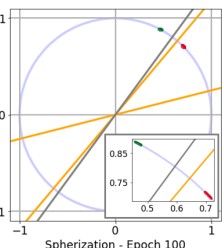

Figure 3: Visualization of Hyperplanes, Decision Boundary, and Representations in the Toy Task. (w: hyperplanes, boundary: decision boundary, red or green points: representations for label 0 or 1)

Table 1: Retention of the Training Ability on Image Classification with Various Datasets and Models (Accuarcy(%): $\mu \pm \sigma$)

| Network | Dataset | Reference | | Reproduced | | Spherized | |
|---|---|---|---|---|---|---|---|
| | | train | test | train | test | train | test |
| SimpleFNN [7] | MNIST | – | 98.47 | 99.99±0.01 | 98.58±0.03 | 99.99±0.01 | **98.65±0.04** |
| LeNet-5 [8] | | – | 99.05 | 99.55±0.09 | 99.10±0.05 | 99.79±0.09 | **99.14±0.04** |
| | F-MNIST | – | 94.70 | 99.24±0.18 | **94.36±0.06** | 98.92±0.36 | 94.34±0.17 |
| VGG-11 [27] | CIFAR10 | – | 90.90 | 100.00±0.00 | 92.38±0.06 | 100.00±0.00 | **92.49±0.11** |
| | CIFAR100 | – | 66.80 | 99.71±0.03 | 68.42±0.12 | 99.82±0.02 | **69.03±0.24** |

Table 2: Retention of Training Ability on Image Classification with CIFAR100 in Various Network Width and Depth Settings (Accuarcy(%): $\mu \pm \sigma$)

| Depth | Width | Reference[2] | | Reproduced | | Spherized | |
|---|---|---|---|---|---|---|---|
| | | train | test | train | test | train | test |
| | 16/32/64/128 | – | – | 79.23±5.94 | 60.17±0.21 | 77.48±5.60 | **60.40±0.35** |
| | 32/64/128/256 | – | – | 98.34±0.37 | 64.89±0.38 | 96.98±3.67 | **65.38±0.28** |
| VGG-11 | 64/128/256/512 | – | – | 99.71±0.03 | 68.42±0.12 | 99.82±0.02 | **69.03±0.24** |
| | 128/256/512/1024 | – | – | 99.90±0.00 | 70.53±0.35 | 99.93±0.00 | **70.89±0.19** |
| | 256/512/1024/1024 | – | – | 99.90±0.01 | 71.43±0.22 | 99.93±0.01 | **71.94±0.19** |
| VGG-11 | | – | 68.64 | 99.71±0.03 | 68.42±0.12 | 99.82±0.02 | **69.03±0.24** |
| VGG-16 | 64/128/256/512 | – | 72.93 | 99.39±0.07 | 72.51±0.26 | 99.54±0.05 | **72.53±0.17** |
| VGG-19 | | – | 72.23 | 97.95±0.81 | 71.53±0.32 | 99.30±0.06 | **72.17±0.33** |

**Hyperplanes**    Hyperplanes and decision boundary in the feature space are defined as follows with the parameters of the output layer:

$$\mathcal{W}_1 : \mathbf{w}_1 \cdot \mathbf{x} + b_1 = 0$$
$$\mathcal{W}_2 : \mathbf{w}_2 \cdot \mathbf{x} + b_2 = 0$$
$$\mathcal{D}_{12} : (\mathbf{w}_1 - \mathbf{w}_2) \cdot \mathbf{x} + (b_1 - b_2) = 0$$

, where $\mathcal{W}_i$ is the hyperplane determined by weight parameter $\mathbf{w}_i$ and bias $b_i$, and $\mathcal{D}_{ij}$ is a linear decision boundary whose points satisfy 0.5 confidence for both labels. In the spherized network, all bias terms are eliminated. In Figure 3, the hyperplanes are illustrated as yellow lines and the decision boundary as a gray line. In the spherized network, hyperplanes and decision boundary pass through the origin from the initial to the last epoch, unlike the original network. The results imply that the spherization layer can learn the correct decision boundary by changing only the angles of hyperplanes passing through the origin.

### 4.2    Retention of Training Ability on Image Classification Benchmarks

**Implementation Details**    In this task, we empirically verified that replacing an existing layer to a spherization layer still maintains the training ability of the original networks on well-known image classification tasks. We reproduced all networks and their performance on the image classification tasks with each dataset. Then, we validated our proposed method on the same settings with them, where only the last fully connected layer replaced by the spherization layer. See Appendix C for the detail about networks and datasets.

Table 3: Analysis on the Effect of Projection on Image Classification with CIFAR10 (acc.(%): accuracy, # err.: the number of errors in overlapping samples, # ovlp.: the number of overlapping samples, ratio(%): ratio of # err. to # ovlp.) (the number of test data = 10000)

| Role | Operator | No-bias | Train | Test | | | |
|------|----------|---------|-------|------|---|---|---|
| | | | acc. | acc. | # err. | # ovlp. | (ratio) |
| base | Original Conv. | | 99.47 ± 0.42 | 92.46 ± 0.10 | 0 ± 0 | 0 ± 0 | ( 0.00 ± 0.00 %) |
| direct | Sigmoid | | 81.74 ± 4.48 | 79.03 ± 2.71 | 1097 ± 266 | 5222 ± 391 | (20.75 ± 3.70 %) |
| | Linear | | 74.71 ± 1.06 | 72.15 ± 0.32 | 1631 ± 39 | 6849 ± 286 | (23.84 ± 0.63 %) |
| | Cosine | | 77.41 ± 0.89 | 76.15 ± 0.86 | 776 ± 132 | 3170 ± 330 | (24.39 ± 2.66 %) |
| indirect | SW-Softmax | ✓ | 98.32 ± 0.07 | 91.51 ± 0.19 | 167 ± 9 | 7925 ± 58 | ( 2.11 ± 0.11 %) |
| | LW-Softmax | ✓ | 86.74 ± 4.06 | 82.12 ± 3.82 | 946 ± 374 | 7669 ± 66 | (12.30 ± 4.84 %) |
| | CW-Softmax | ✓ | 99.66 ± 0.04 | 92.29 ± 0.18 | 80 ± 5 | 7051 ± 134 | ( 1.14 ± 0.05 %) |
| proposed | Spherization | ✓ | 99.66 ± 0.05 | 92.38 ± 0.14 | 0 ± 0 | 499 ± 106 | ( 0.00 ± 0.00 %) |

**Results and Analysis**  The accuracy results of original and spherized networks are compared in Table 1. The reference and reproduced results of each setting are similar. The accuracy results of the spherized networks implemented on the reproduced code are similar or slightly higher on both training and test data than those of the original networks. These results imply that the spherization layer maintains the training ability of the original network. In Table 2, the accuracy results of original and spherized networks are shown in various width and depth settings. The spherized network consistently exhibits similar or higher test accuracy over all width and depth settings. The results reveal that the spherization layer again preserves the training ability of the original network.

### 4.3   Analysis: Effect of Projection

**Implementation Details**  We compared the method with another angle-based approach to analyze the effect of reducing information loss by projection. For comparison, we used 9-layer CNN, namely CNN-9, with the same experimental setup of [16] but reproduced in PyTorch. To apply a spherization layer, we changed the last fully connected layer in each model. We considered samples overlapped when their cosine similarity is greater than or equal to $(1 - 10^{-6})$.

**Results and Analysis**  After training, we extracted representations from the target layer and analyzed the effect of projection. First, we count the number of representations overlapping at least one another representation. Then, we get the total errors caused by the incorrectly classified and overlapping representations. Finally, we calculate the ratio of the incorrectly classified representations to the overlapping representations. The results of them are shown in Table 3: *# ovlp.*, *# err.*, and *(ratio)*, respectively. As the result, large proportion of representations suffer the overlap problem from projection approach, and most significant errors are caused by the overlapping representations. Furthermore, representation learning relying on bias parameters (direct projection case) causes more significant errors than no-bias layer (indirection projection case) because the representations are more difficult to be distinguished than those of no-bias layer. In comparison, the spherization layer removes the errors and decreases the upperbound of errors measured by the number of overlapping representations.

### 4.4   Analysis: Gradient Flows

**Implementation Details**  The replacement of a hidden layer and the operations in the spherization layer, such as angularization function and conversion-to-Cartesian, might make the gradient flow unstable in the original networks. To empirically verify how the spherization layer affects to the gradients during training, we qualitatively compare the gradient flows of original and spherized VGG-11. They were trained on the image classification with CIFAR100. We used the same setting with Section 4.2.

**Results and Analysis**  As shown in Figure 4, the average of absolute gradients in the original and spherized network are not very different during the whole training. Furthermore, the results show similar flows not only at the last fully connected layer (`fc3` or `sph_fc3`) but at the previous layers. This result implies the spherization layer does not destroy the gradient flows.

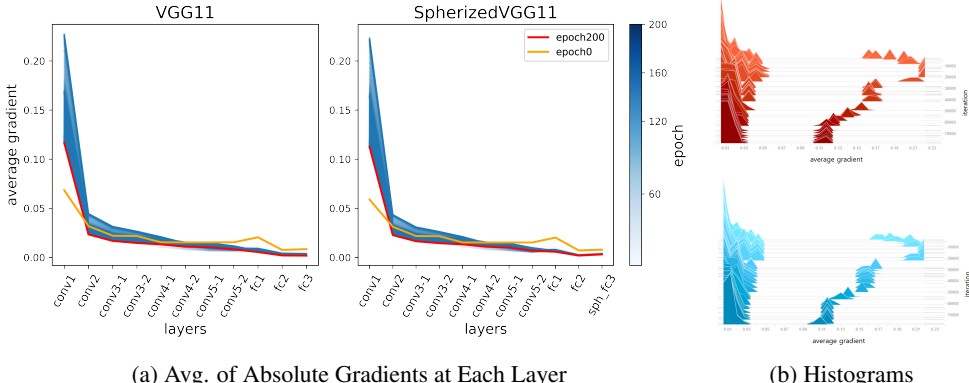

(a) Avg. of Absolute Gradients at Each Layer        (b) Histograms

Figure 4: Analysis of Gradient Flow from Image Classification Model trained on CIFAR100. (a) The y-axis means the average of absolute gradients which occurred at each layer. The left side shows the gradient flow in VGG-11 (VGG11), and the right side shows the spherized VGG-11 (SpherizedVGG11), where the last fully connected layer is substituted with the spherization layer. (b) The histograms show the frequency of the average of absolute gradients in VGG-11 (red) and the spherized VGG-11 (cyan), respectively.

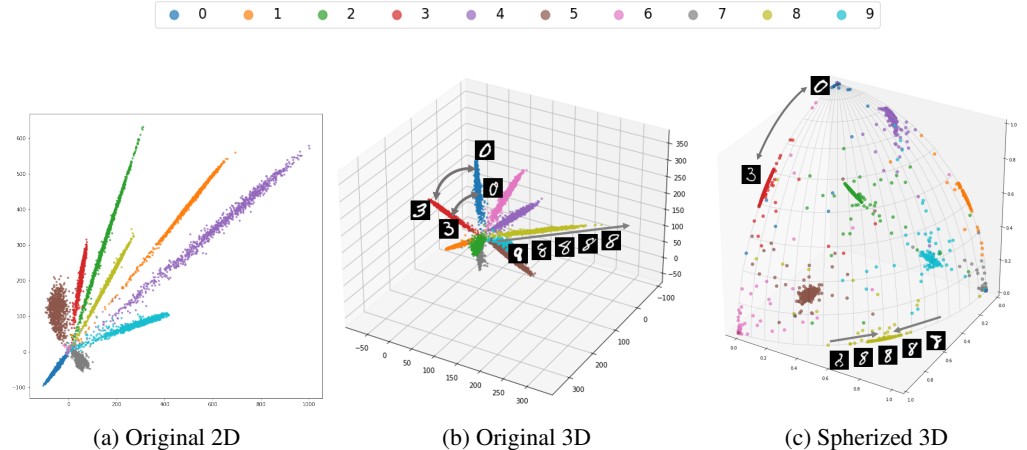

(a) Original 2D        (b) Original 3D        (c) Spherized 3D

Figure 5: Visualization of Feature Representations on MNIST. (a) and (b) are the visualization results of 2D and 3D feature vectors in the original networks, and (c) is the result in the spherized network

## 4.5 Downstream Tasks: Visualization

**Implementation Details**    To investigate the spherization layer locates feature vectors on the $(n + 1)$-spherical surface, we implemented a simple CNN to learn 3-dimensional feature vectors for visualization. This technique is called CNN-Vis3D, which generates 3-dimensional feature vectors and shows those representations as a graph. Detailed configurations are presented in Appendix C. In the spherized network, the second fully connected layer was replaced by the spherization layer, which is located at the previous of the last fully connected layer. After training, the pre-activations before the no-bias layer were used to visualization.

**Results and Analysis**    The results of visualization are illustrated in Figure 5. In the 2D and 3D feature visualization results of the original network, the representations are distributed over wide ranges of both scale and angles, as shown in Figure 5a and 5b. As illustrated in decoupled network [12], the angle accounts for semantic difference and the Euclidean norm accounts for intra-class variation. In the spherized network, all representations are placed on the 3-spherical surface, as shown in Figure 5c. Based on these spherized representations and the following no-bias layer, all the roles are expressed by the angle change on the hyperspherical surface. Thus, in the spherization layer, all trained information can be used in angular similarity-based interpretation.

Table 4: Performance on the Word Analogy Test ($S_{ppl}/S_{pmi}/S_{mppl}$ ↑)

| Model | SAT | U2 | U4 | Google | BATS | Avg. |
|---|---|---|---|---|---|---|
| BERT | **29.4**/28.5/**28.8** | 36.0/36.0/**36.8** | **38.7**/34.7/34.3 | **33.0**/**33.8**/**33.0** | 32.3/**35.0**/33.2 | **33.9**/33.6/**33.2** |
| BERT + *sph* | 29.1/**29.4**/27.9 | **37.3**/**39.0**/36.0 | 36.8/**35.9**/**35.4** | 32.4/32.6/32.2 | **34.0**/34.2/**33.8** | **33.9**/**34.2**/33.1 |
| RoBERTa | **29.4**/31.2/29.7 | 35.5/**35.5**/**36.4** | 33.6/**34.3**/**34.5** | 32.8/**33.2**/30.8 | 30.9/**31.6**/30.9 | 32.4/**33.1**/32.5 |
| RoBERTa + *sph* | 29.1/29.4/**30.0** | **36.4**/**35.5**/34.2 | **34.0**/**34.3**/33.3 | **34.2**/**33.6**/**32.8** | **35.0**/**33.9**/**34.8** | **33.7**/**33.3**/**33.0** |

Table 5: Performance of few-shot learning on Mini-ImageNet. Euclidean and Cosine mean euclidean distance and cosine similarity, respectively, which are the distance metrics that used in the experiments. (Accuarcy(%): $\mu \pm \sigma$)

| Model | Test Acc. | | Model | Test Acc. | |
|---|---|---|---|---|---|
| | Euclidean | Cosine | | Euclidean | Cosine |
| ConvNet | 50.29±0.18 | 52.87±0.18 | ResNet | 37.63±0.15 | 33.41±0.15 |
| ConvNet + *sph* | 43.41±0.16 | **53.74±0.16** | ResNet + *sph* | 31.77±0.13 | **38.71±0.16** |

### 4.6 Downstream Tasks: Word Analogy Test

**Implementation Details**   We used BERT [4] and RoBERTa [17] to conduct the word analogy test, in which angular similarity is used to predict a relation type between words. The settings were identical with [25]. We applied the spherization layer to the last fully connected layer of the encoder in each model. Next, we trained them on WikiText [19] for 3 epochs with 8 mini-batches, the softmax function following cross-entropy, SGD at a learning rate 0.0001 on masked-language modeling.

**Results and Analysis**   Table 4 presents the performance evaluated with three metrics [25]. Applying the spherization layer to BERT and RoBERTa improves most average scores. The improvement implies that the spherized representations provide accurate information for angle-based distinction of word relations.

### 4.7 Downstream Tasks: Few-shot Learning

**Implementation Details**   We used ProtoNet [24] with ConvNet and ResNet for few-shot learning on Mini-ImageNet [26], in which several feature vectors are compared with the feature vector of an input image by using distance metric. This task was performed according to the guidelines in [2].

**Results and Analysis**   Table 5 details the performance results in few-shot learning [2]. Generally, the Euclidean distance is used to discriminate feature vectors. However, the Eucliean distance also has the dispersion problem. To focus on only the angles, we trained the models with cosine similarity and compared the performances. As shown in Table 5, all spherized models with cosine similarity outperform the other models. This improvement indicates the spherized representations are useful for the angle-based metric such as cosine similarity.

## 5 Related Works

**Semantic Analysis on the Inner Product**   The inner product is a crucial operator in current neural networks, in which the distance between input vector **x** and weight vector **w** is encoded. At the decoupled networks [12], the inner product is reparametrized with the norms and the angle, and the intra-class variation and the semantic difference are modeled in neural networks by decoupling them. Furthermore, the substitutes of the inner product have been proposed, where the direction of gradient or the similarity between kernels are used as the key factor instead of the inner product [11, 31]. The spherization layer can be considered to be the substitute of the inner product, which normalizes the input vectors by locating them on the hyperspherical surface. However, the spherization layer is a direct and specific method to convert feature vectors focused on the angles without information loss.

**Angle-based Approach**   The semantic analysis on the inner product has revealed that the streams focuses on the information in the angles. The angle is a crucial factor, in which the most abundant and discriminative information is preserved [1, 9, 10, 15]. In SphereFace variants [10, 13, 14, 32],

the angular softmax that enables CNNs to learn discriminative features on angular separability was used. Furthermore, some of this angular information have been used for regularization [29, 30]. In most angle-based studies, the angular information was used indirectly by the objective function or regularization. In contrast, the spherization layer ensures the model directly learns the angular information on the hyperspherical surface.

**Hyperspherical Representation Learning**    In some angle-based approaches, input vectors were directly projected onto the hyperphserical surface [12, 16, 18]. These hyperspherical representation learning methods normalized the input vectors to ensure models are dependent on only the angles. However, this normalization is the projection onto the hyperspherical surface, and it can be less discriminative when some points overlap after the projection. The spherization layer locates the input vectors on the hyperspherical surface without the overlap problem through the spherization.

## 6    Conclusion

We introduced the dispersion problem of trained information to the Euclidean norm and angle on representations. To address the dispersion problem, we proposed the *spherization layer* to learn representations by using only the angles without information loss. We used the angularization for using pre-activations as angular coordinates, conversion-to-Cartesian for locating them on the $(n+1)$-spherical surface, and no-bias training to learn representations by using only the angles. In the experiments on toy, image classification benchmarks, few-shot learning, and word analogy test, the proposed method achieved accurate learning of the decision boundary and retention of the original training ability, and improved performance in downstream tasks using angle-based information re-used or interpreted. The proposed method can be applied to numerous network layers and downstream applications. A limit of this approach is that the spherization layer should be inserted to networks in a training step to fully utilize its advantage, which restricts the use of pre-trained models trained without it. Recovering trained information on representations with sampling should be investigated in the future.

## Acknowledgments and Disclosure of Funding

This work was partially supported by the National Research Foundation of Korea (NRF) grant funded by the Korea government (MSIT) (2022R1A2C2012054), and by Institute of Information & communications Technology Planning & Evaluation (IITP) grant funded by the Korea government (MSIT) (No.2019-0-01842, Artificial Intelligence Graduate School Program (GIST)).

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
