# Spherization Layer: Representation Using Only Angles (w. Supplementary Material)

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

## A Lower bound $\varphi_L$

To prevent the last coordinate from an extremely small value, we set a lower bound $\varphi_L$ of angles. When we assume that the trigonometric value of all angular coordinates is $\alpha$, $\sin \varphi = \alpha$. Based on this assumption, the lower bound $\varphi_L$ can be obtained as illustrated in Eq. 12.

$$
\begin{aligned}
when \ \sin \varphi = \alpha, \ \alpha^n \geq \delta \ &\Leftrightarrow \ \alpha = \delta^{1/n} \\
\varphi = \ \sin^{-1} \alpha \geq \ &\sin^{-1} \left( \delta^{1/n} \right) \\
\therefore \ \varphi_L = \sin^{-1} &\left( \delta^{1/n} \right)
\end{aligned}
\tag{12}
$$

## B Calculation Trick

To convert the angular coordinates $\boldsymbol{\varphi} \in \mathbb{R}^n$ to Cartesian coordinates $\mathbf{s} \in \mathbb{R}^{n+1}$, we first expand the dimension of $\boldsymbol{\varphi}$ by using the constant matrix $\mathbf{W}_\varphi = [\mathbf{I}_n; \mathbf{v}] \in \mathbb{R}^{n \times (n+1)}$, where $\mathbf{v} = [\mathbf{0}; 1]^\top \in \mathbb{R}^n$ and $[\cdot ; \cdot]$ means concatenation. Then, we can get the expanded angular vector $\boldsymbol{\phi}$ as illustrated in Eq. 8.

When we define $\mathbf{s}'$ as illustrated in Eq. 13 from Eq. 8, we calculate the $i$-th coordinate $s_i$ as shown in Eq. 14 , where $\mathbf{b}_\phi = [\mathbf{0}; -\frac{\pi}{2}]^\top \in \mathbb{R}^{n+1}$ and $\mathbf{W}_\phi \in \mathbb{R}^{(n+1) \times (n+1)}$ is an upper triangular matrix in which all diagonals are zero, $(\mathbf{W}_\phi)_{n,n+1}$ is also zero, and the other elements are ones.

$$
\mathbf{s}' = \mathbf{W}_\phi^\top \ln (\sin \boldsymbol{\phi}) + \ln \left( \cos \left( \boldsymbol{\phi} + \mathbf{b}_\phi \right) \right)
\tag{13}
$$

$$
\begin{aligned}
s'_{1 < k < n+1} &= \sum_{i=1}^{k-1} \ln (\sin \varphi_i) + \ln (\cos \varphi_k) \\
&= \ln \left( \cos \varphi_k \prod_{i=1}^{k-1} \sin \varphi_i \right) \\
s'_1 &= \ln (\cos \varphi_1) \\
s'_{n+1} &= \ln \left( \prod_{i=1}^{n} \sin \varphi_i \right)
\end{aligned}
\tag{14}
$$

Finally, we get the Cartesian coordinates $\mathbf{s}$ from this calculation trick as illustrated in Eq. 15.

$$
\begin{aligned}
\mathbf{s} &= r \cdot \exp(\mathbf{s}') \\
&= [r \cos \varphi_1, \cdots, r \cos \varphi_k \prod_{i=1}^{k-1} \sin \varphi_i, \cdots, r \prod_{i=1}^{n} \sin \varphi_i]
\end{aligned}
\tag{15}
$$

## C Implementation Details

**Image Classification** We conducted image classification experiments on MNIST [4], Fashion-MNIST [40], CIFAR10, and CIFAR100 [10]. For MNIST, we used a 3-layer neural network [8] and LeNet-5 [11]. We followed the settings in [8] and in [11], and the architectures of them are illustrated in Table 6 SimpleFNN and LeNet-5, respectively. Unlike [8], we used 256 mini-batches, Adam at a learning rate of 0.001 and weight decay of 5e-5 at the 3-layer neural network. At LeNet-5, we set 128 mini-batches, Adam at a learning rate of 0.001, and the other settings are same with [11]. For Fashion-MNIST, CIFAR10, and CIFAR100, we used VGG variants [28] with batch normalization [9] denoted as VGG-*N*. ReLU, 128 mini-batches, and SGD with momentum of 0.9 and with weight decay of 5e-4 were used as default in three datasets. The number of epochs was set to 100, 300, and 200, respectively. For Fashion-MNIST, the learning rate was initially 0.01 and it was divided by 2 at every 20 epochs. For CIFAR10, we set the initial learning rate as 0.05 and reduced it to half at every 30 epochs. For CIFAR100, the learning rate was initially 0.1 and divided by 5 at 60th, 120th, and 160th epochs. The last fully connected layer of the backbone network was replaced by the spherization layer for each task. As pre-activations used for generating angular coordinates, the range of angles was reduced when dropout. To resolve this problem, we removed the dropout in front of the spherization layer. For analysis of robustness of the spherization layer to width and depth

Table 6: Network configurations. The convolutional layer parameters are denoted as "conv(receptive field size)-(number of channels)", and the ReLU activation function is not shown for brevity [28]. The fully connected layer parameters are denoted as "FC-(number of neurons)". In CNN-Vis3D, $N$ is set as 2 or 3 to get the feature vectors for visualization. In VGG-11 ($C$), $C = [C_1, C_2, C_3, C_4]$ means the number of channels at each convolutional block. $N_{out}$ is set as the number of classes in each image classification task. The highlighted layer means where it is replaced by the spherization layer in the spherized network.

| SimpleFNN | LeNet-5 | CNN-Vis3D | CNN-9 | VGG-11 ($C$) | VGG-16 | VGG-19 |
|---|---|---|---|---|---|---|
| FC-500 | conv5-6 | conv3-32 | conv3-128 | conv3-$C_1$ | conv3-64 | conv3-64 |
| FC-300 | conv5-16 | conv3-32 | conv3-128 | | conv3-64 | conv3-64 |
| **FC-10** | FC-120 | maxpool | conv3-128 | maxpool | | |
| soft-max | FC-84 | conv3-64 | maxpool | conv3-$C_2$ | conv3-128 | conv3-128 |
| | **FC-10** | conv3-64 | conv3-192 | | conv3-128 | conv3-128 |
| | soft-max | maxpool | conv3-192 | maxpool | | |
| | | conv3-128 | conv3-192 | conv3-$C_3$ | conv3-256 | conv3-256 |
| | | conv3-128 | maxpool | conv3-$C_3$ | conv3-256 | conv3-256 |
| | | FC-256 | conv3-256 | | conv3-256 | conv3-256 |
| | | **FC-$N$** | conv3-256 | | | conv3-256 |
| | | FC-10 | conv3-256 | maxpool | | |
| | | soft-max | maxpool | conv3-$C_4$ | conv3-512 | conv3-512 |
| | | | FC-256 | conv3-$C_4$ | conv3-512 | conv3-512 |
| | | | **FC-10** | | conv3-512 | conv3-512 |
| | | | soft-max | | | conv3-512 |
| | | | | maxpool | | |
| | | | | conv3-$C_4$ | conv3-512 | conv3-512 |
| | | | | conv3-$C_4$ | conv3-512 | conv3-512 |
| | | | | | conv3-512 | conv3-512 |
| | | | | | | conv3-512 |
| | | | | maxpool | | |
| | | | | FC-4096 | | |
| | | | | FC-4096 | | |
| | | | | **FC-$N_{out}$** | | |
| | | | | soft-max | | |

configurations, we used the VGG backbone again and changed the convolutional configuration of it on CIFAR100. To generate variations on width, we set the number of channels at Conv1.x, Conv2.x, Conv3.x, and Conv4.x to 16/32/64/128, 32/64/128/256, 64/128/256/512, 128/256/512/1024 and 256/512/1024/1024, respectively. The filter in Conv5.x has same number with the filter in Conv4.x (see Table 6). To give variations on depth, we used VGG-11, VGG-16, and VGG-19.

**Visualization**  CNN-Vis3D is composed of three convolutional layer blocks whose kernel size are 32, 64, and 128, as shown in Table 6. To obtain 3-dimensional feature vectors, we set the number of neurons at the second fully connected layer, which locates before the last fully connected layer, as 3 in the original network. For a fair comparison, we replaced this layer with the spherization layer and changed the number of neurons to 2, and visualized 2-dimensional feature vectors from the original network, in which the number of neurons at the second fully connected layer is 2. In the CNN-Vis3D for 2D and 3D, named Original 2D and Original 3D respectively, batch normalization, ReLU, 64 mini-batches, cross-entropy, and Adam at a learning rate 0.001 were used as default. Unlike the original network, we set 32 mini-batches in the spherized CNN-Vis3D, called Spherized 3D. We trained original 2D, original 3D, and Spherized 3D for 20 epochs, and they converged and achieved the training accuracies(%) 98.87, 99.48, and 99.13, respectively.

# D   Effect of Fine-tuning

Table 7: Performance on Word Analogy Test (fine-tuning) ($S_{ppl}/S_{pmi}/S_{mppl}$ ↑)

| Model | SAT | U2 | U4 | Google | BATS | Avg. |
|---|---|---|---|---|---|---|
| BERT | **29.7/32.3**/29.4 | **37.3/36.0/34.2** | **39.1/34.7/34.6** | **44.8**/44.4/44.2 | **44.1/41.3/40.7** | **39.0/37.7/36.6** |
| BERT + *sph* | 29.1/29.1/**30.0** | 36.8/35.1/33.3 | 38.2/32.6/33.3 | 44.6/**45.8/44.4** | 40.8/38.8/39.3 | 37.9/36.3/36.1 |
| RoBERTa | 40.7/38.0/40.1 | 43.4/47.8/45.6 | 41.2/43.3/42.1 | 83.0/84.0/83.2 | 68.1/68.5/69.3 | 55.3/56.3/56.0 |
| RoBERTa + *sph* | **41.5/42.1/42.7** | **49.6/50.4/47.8** | **46.3/46.8/46.5** | **88.0/87.0/88.4** | **71.0/70.7/70.9** | **59.3/59.4/59.3** |

Table 8: Performance of Few-shot Learning on Mini-ImageNet (fine-tuning) (Accuarcy(%): $\mu \pm \sigma$)

| Model | Test Acc. | | Model | Test Acc. | |
|---|---|---|---|---|---|
| | Euclidean | Cosine | | Euclidean | Cosine |
| ConvNet | **67.80±0.17** | 66.60±0.17 | ResNet | 77.73±0.15 | **78.86±0.15** |
| ConvNet + *sph* | 51.56±0.17 | 61.77±0.17 | ResNet + *sph* | 72.18±0.15 | 74.57±0.15 |

In neural networks, the fine-tuning technique is used in natural language processing [25, 39] and computer vision [30, 41]. In this method, the pre-trained model is used and re-trained on a new target domain. Fine-tuning can achieve excellent performance. However, for spherization, a layer in the architecture should be replaced with the spherization layer, and this spherized network should be trained from scratch. If not, a mismatch between the learned representations and spherization unexpectedly affects the performance of the fine-tuned model. To verify this phenomenon, we investigated fine-tuning on the word analogy test and few-shot learning. We used the pre-trained weights from [39] for BERT and RoBERTa, and from [2] for ConvNet and ResNet. For synchronizing with spherized features, we loaded the pre-trained models and fine-tuned them on WikiText dataset [22] and Mini-ImageNet [32], respectively. See the experimental details and results on following subsections.

## D.1   Word Analogy Test

**Implementation Details**   Unlike the few-shot learning, there are no data to re-train the pre-trained weights in word analogy test because they use language models to evaluate the performance on word analogy test. To resolve this problem, we used WikiText [22] to make fine-tuned language models. We loaded pre-trained weights BERT and RoBERTa, respectively, and trained them on WikiText for 3 epochs, 8 mini-batches, softmax function following cross-entropy, SGD at a learning rate 0.0001. For spherization, we replaced the last fully connected layer of encoder in each network with the spherization layer.

**Result and Analysis**   The results of fine-tuning on word analogy test are illustrated in Table 7. In the original networks, the performances of both are obviously increased. The spherized BERT also have improvements but not much as the original network, while the spherized RoBERTa shows greater improvement. This result means that the angular information in the pre-trained representations might be more helpful enough to overcome the information collapse.

## D.2   Few-shot Learning

**Implementation Details**   The implementation details were used according to the method presented in Section 4.7. For fine-tuning, we obtained the pre-trained weights [2] for ConvNet and ResNet, respectively. In the spherized networks, the part of last fully connected layer was substituted with the spherization layer when loading these weights. Thus, the pre-trained weights of this layer were not necessary anymore.

**Result and Analysis**   The results of fine-tuning on few-shot learning are shown in Table 8. In the original networks, the performances increase considerably. The spherized networks exhibit improvements but not as much as the original networks. This result indicates that there are some information collapse when training the spherization layer on pre-trained representations but it is not that fine-tuning is not helpful at all. To the best of our knowledge, the angular information in the pre-trained representations might be helpful to train spherized representations.