# OpenReview forum: "Spherization Layer: Representation Using Only Angles"
_NeurIPS.cc/2022/Conference — NeurIPS 2022 Accept_

### Official Review · Reviewer_v1RE · 2022-07-09

**Rating:** 5
**Confidence:** 2
**Soundness:** 3 good
**Presentation:** 3 good
**Contribution:** 3 good

**Summary:**

The authors propose a novel neural network layer that forces pre-activations into angles. Besides, the spherization Layer converts the angular coordinates to Cartesian coordinates and the training is done without bias parameters. Several different empirical experiments have shown the effectiveness of the proposed method.

**Questions:**

It would be nice if the authors can add more explanation about the motivation and analysis of the conversion-to-Cartesian process.

**Limitations:**

The authors have adequately addressed the limitations of this paper.

**Strengths And Weaknesses:**

Strengths:

1. This paper is well written and clear.
2. The proposed spherization layer can be easily applied to most existing neural network architectures and downstream tasks.

Weakness:

1. There is not enough comparison between the proposed spherization layer and other recent angle-based learning methods such as [1] [2]. Since the authors claim that the proposed method solves the information loss problem that occurs in these projection-based learning methods, these comparisons are important experiments to support the authors claims.


[1] Chen, Beidi, et al. "Angular visual hardness." International Conference on Machine Learning. PMLR, 2020.
[2] Lin, Rongmei, et al. "Regularizing neural networks via minimizing hyperspherical energy." Proceedings of the IEEE/CVF Conference on Computer Vision and Pattern Recognition. 2020.

---

> ### Author Response · Authors · 2022-08-02
> **Explain the reason why an explict empirical comparison with [1] and [2] were excluded and add more explanation about the motivation and analysis of the conversion-to-Cartesian process**
>
> Thank you for your valuable comments. We used notations W1 for the weakness, and Q1 for the question in our response.
>
> **W1** We agree that the papers and our method commonly use angle-based learning methods. However, [1] and [2] you mentioned have used the angular information for regularization. We want to point out that our motivation is to solve the information loss by projection while the works are to obtain better inductive bias, not to learn representations dependent only on the angles. Because the causes of the two effects are different from ours, we excluded them from direct empirical comparison.
>
> **Q1** In the conversion-to-Cartesian stage, the angular coordinates, which come from the angularization that converts the pre-activations to angular coordinates, are converted to Cartesian coordinates on the (n+1)-spherical surface.
>
> *[motivation]* Without this conversion, a following no-bias layer is impossible to be trained on the output of angularization because the coordinate system of the output is different from the input of the layer. For this reason, we ensure the consistency between the output of angularization (polar coordinate system) and the input of the following no-bias layer (Cartesian coordinate system) through the conversion-to-Cartesian. As a result, this conversion enables the layer to be trained in the same way as general neural networks.
>
> *[analysis]* One more, it should be discussed whether the spherization layer has enough capacity to be compatible with a Euclidean layer after the conversion-to-Cartesian. As the result of conversion-to-Cartesian, feature representations are located on (n+1)-spherical surface, where the dimension is one larger than a Euclidean layer, and these representations are fed to the following no-bias layer, which is possible to be trained by using only the angles. Therefore, the spherization layer, more precisely the no-bias layer, has the same number of weight parameters as the Euclidean layer. In terms of geometry, the relation between the surface of (n+1)-sphere $S_{n+1}$ and the volume of n-sphere $V_n$ is $S_{n+1}=2 \\pi V_n$. So, the spherization layer can cover the search space as enough as a Euclidean layer does.

---

### Official Review · Reviewer_B9zn · 2022-07-10

**Rating:** 7
**Confidence:** 4
**Soundness:** 3 good
**Presentation:** 3 good
**Contribution:** 3 good

**Summary:**

This paper proposes spherization layer to represent all information on angular similarity, which avoids the information loss when using only angular similarity on representations trained with the inner product. It maps the pre-activations of input vectors into the specific range of angles, converts the angular coordinates of the vectors to Cartesian coordinates with an additional dimension, and trains decision boundaries from hyperplanes. This guarantees that representation learning always occurs on the hyperspherical surface without the loss of any information unlike other projection-based methods. Experiments have been evaluated on different tasks and show the effectiveness of the proposed methods.



**Questions:**


I have no problem

**Limitations:**

Weakness.
Some related works should be listed and have a discussion and comparison,

**Strengths And Weaknesses:**

Novelty.
This paper proposes a novel method for representation learning only using the angle information. The subtle construction to covert the angular coordinates of the vectors to Cartesian coordinates, which avoids the information loss than the previous method.

Writing.
This paper is of better organized and shows detailed description about the proposed method.

---

> ### Author Response · Authors · 2022-08-02
> **Try to organize better, and add more explanations and details**
>
> Thank you for your valuable comments. We used a notation W1 for the weakness in our response.
>
> **W1** To enhance the point, we reorganized some contents and added more explanations and details as listed below.
>
> 1. (method) Motivation for conversion-to-Cartesian process in Section 3
> 2. (related work) More works about angle-based learnings in Appendix E
> 3. (detailed description) Clear description of the title of Table 2
> 4. (detailed description) Analyze the gradient flow of our proposed method in Appendix F
> 5. (better organized) Reorganize Section 4.3 to clarify our intention
> 6. (better organized) Re-arrange the tables and figures to make our paper more readable with explanations about them.

---

### Official Review · Reviewer_ouKB · 2022-07-11

**Rating:** 7
**Confidence:** 3
**Soundness:** 4 excellent
**Presentation:** 3 good
**Contribution:** 3 good

**Summary:**

The paper proposes a spherization layer to represent preactivations in a hypersphere. This layer can be plugged into any neural network. This representation is useful in cases where cosine similarity is used as the main metric to determine similarity in the feature space.

**Questions:**

1. It seems it is possible to theoretically show that the spherization layer retains all information as a Euclidean layer. Meaning, a spherization layer can learn any function that can be represented by a Euclidean layer. Can you comment on this?

**Limitations:**

Societal impact is not discussed.

**Strengths And Weaknesses:**

## Strengths
1. The spherization idea is interesting. It addresses a niche problem in a neat way.
2. The formulation is clean and clear justifications are provided.
3. Experiments are performed in various settings and significant improvements are observed in the few-shot learning setting.

## Weaknesses
1. The improvement is not clear compared to the projection approach (table 5). This raises doubts about the motivation for this approach. Is there really an information loss with the projection approach?
2. More experiments where explicit similarity is used in the feature space could be performed. Examples include self-supervised learning and metric learning. This may show the benefits of the proposed approach in a better way.
3. Training stability of the network could have been discussed. It is not clear, how the signal propagation (gradient flow) would be affected by the spherization layer.

## Post rebuttal
Thanks for the clarifications. I'm in favour of accepting the paper. Please consider addressing the discussed comments in camera-ready or in an extension.

---

> ### Author Response · Authors · 2022-08-02
> **Clarify our motivation and show the information loss with the projection approach, and comment on that spherization layer retains all information as a Euclidean layer**
>
> Thank you for your valuable comments. We used notations W1, W2, and W3 for the weaknesses, and Q1 for the question in our response.
>
> **W1** To alleviate this weak point, we tried to give more information about the experiments used in Table 3, as listed below.
>
> 1. The number of representations overlapped at least one another.
> 2. The total errors caused by the incorrectly classified and overlapped representations.
> 3. The ratio of the errors to the overlapped representations.
> 4. The condition of using a no-bias layer
>
> The updated table of Table 3 is shown in Section 4.3, and the new table that contains the above information is shown in Table 9 (Appendix E).
>
> In summary, Table 9 figures out the following points.
>
> 1. Large proportion of representations suffer from projection.
> 2. Considerable errors occurred in the overlapped representations.
> 3. Learning representations relying on bias parameters (direct projection case) causes larger errors than a no-bias layer (indirection projection case).
> 4. Spherization layer considerably reduces the number of overlapped representations
> 5. Spherization layer still leaves a small number of overlapped representations, but they do not cause errors. (We expect that it is caused by the computational error with floating point variables.)
>
> **W2** We agree with the comment. Beyond the experiments on few-shot learning and word analogy test in our paper, other self-supervised learning and metric learning can also be affected by our method. Because of our limited computing resources, we have not covered more experiments for now, but we plan to extend them in future work.
>
> **W3** We appreciate for raising the issue. We also believe that training stability is an important issue. So, we added the analysis on the gradient flow, as shown in Figure 5 (because of the page limit, we put it onto the supplementary) (Appendix F). In summary, the results do not show a significant difference in gradient scales before and after applying the spherization layer.
>
> **Q1** As we understood, the question is whether the spherization layer has enough capacity to be compatible with a Euclidean layer, and the short answer is yes. Through the spherization, feature representations are located on (n+1)-spherical surface (as the result of conversion-to-Cartesian), where the dimension is one larger than a Euclidean layer, and these representations are fed to the following no-bias layer, which is possible to be trained by using only the angles. Therefore, the spherization layer, more precisely the no-bias layer, has the same number of weight parameters as the Euclidean layer. In terms of geometry, the relation between the surface of (n+1)-sphere $S_{n+1}$ and the volume of n-sphere $V_n$ is $S_{n+1}=2 \\pi V_n$. So, the spherization layer can cover the search space as enough as a Euclidean layer does.

---

> > ### Comment · Reviewer_ouKB · 2022-08-08
> > **Some clarifying questions**
> >
> > Thanks for addressing the points. I've a couple of clarifying questions.
> >
> > 1. How are "overlapping representations defined"? There needs to be a threshold to on the distance between two features to say they are overlapped. Also, the point here is that the number of overlapping representations give a measure of information loss?
> > 2. Any intuition on why the average gradient increase with the epochs? I would expect it to decrease as the training progresses. The effect is higher for spherization layer. Can you comment on this?

---

> > > ### Author Response · Authors · 2022-08-09
> > > **Definition of overlapping, clarification of the point, and comment on gradient flow**
> > >
> > > Thank you for your clarifying comments. We used notations Q1 and Q2 fot the questions in our response.
> > >
> > > **Q1**
> > >
> > > *[definition of overlapping]* We first got pair-wise cosine similarity between representations. Then, we regarded the points overlapped when there is at least one representation that the cosine similarity is higher than or equal to a threshold 1 - $\\delta$, where $\\delta=1e^{-6}$.
> > >
> > > *[clarification of the point]* We revised the sentence in Appendix E to clarify this point: "the spherization layer reduces the number of overlapped representations" -> "the spherization layer removes the errors and decreases the upperbound of errors measured by the number of overlapping representations". Rather than the number of overlapping representations, the number of errors in overlapping representations gives the measure of information loss. More precisely, in Table 9, we checked the performance degradation as the number of errors in overlapping representations (*# err.*) increased. In this observation, the spherization removed the errors and decreased the upperbound of errors measured by the number of overlapping representations.
> > >
> > > **Q2** To the best of our knowledge, an increase or decrease in the average of absolute gradients depends on the shape of the loss landscape. Especially, when the convergence point is located on the oscillated landscape, the average gradient keeps a large scale until the end of training. To show the opposite case (i.e., the average gradient decrease with the epochs), we observed the gradient flow in VGG-11 on image classification with CIFAR100 as shown in Figure 6 (Appendix F), and the average gradient decreased with the epochs. Also, the effect depends on the environment, not the spherization layer. The gradient flow in VGG-11 supports this fact by empirically showing that the spherization layer does not make the average gradient higher.

---

> > > > ### Comment · Reviewer_ouKB · 2022-08-09
> > > > **Minor things**
> > > >
> > > > Thank you for the clarification. Two minor things, possibly in an extension.
> > > >
> > > > 1. It would be great to have a principled measure of information loss and compare methods.
> > > > 2. It would be nice to rigorously prove that the spherization layer can cover the search space as enough as a Euclidean layer does.
> > > >
> > > > Having said that, these should not be a reason for rejecting this paper.

---

### Official Review · Reviewer_Wemx · 2022-07-11

**Rating:** 7
**Confidence:** 3
**Soundness:** 3 good
**Presentation:** 3 good
**Contribution:** 3 good

**Summary:**

This paper presents a new formulation for a layer in neural networks known as Spherization Layer. When inner product with Euclidean norm is calculated, often there exist a loss of scalar information due to normalization. Authors propose Spherization layer to form a bijective mapping from Cartesian coordinates to hyperspherical surface. Spherization layer can replace typical neural network layers. Empirical experiments, from simple synthetic data to real-world dataset such as MNIST, Mini-ImageNet and WikiText, show that performance of neural network with Spherization layer is scalable and can perform well.

**Questions:**

1. Although authors motivate the formulation in the beginning, stating that scalar information can be lost. What is that information exactly? Is it possible to show empirically, either via a small experiment or via real-world experiment, to show formulations that uses normalization loses information due to scaling?

2. Can authors further elaborate on what you mean by ”Robustness” in Figure 2? Robustness at first glance might remind some readers of adversarial robustness, or robustness to noise or some deformation on the input. Upon further reading, it seems to mean “maintaining trainability.” I recommend changing the wording here to avoid confusion, or easier readability.

**Limitations:**

Authors stated potential limitations of their work, in that spherization layer only takes on an effect if its used during training. It is mentioned that it will require retraining in order to use the proposed formulation in conventional pretrained models.

**Strengths And Weaknesses:**

The formulation presented is clear and well-written. Background on spherical coordinates in $n+1$ dimensions, and step-by-step explanation of the spherization process is easy to follow with good choice of notation. Experiments are also well-documented. The toy example is a good starting point verifying the formulation. Experiments on both computer vision tasks such as image classification and language tasks such as word analogy are also pluses that show case formulation proposed is versatile for different use cases. Empirically, for some setting, proposed formulation can greatly outperform older related works. The formulation is original, and significant in that it is a flexible formulation and is applicable in many other related works.

I reserve the weakness and questions together for better coherence.

---

> ### Author Response · Authors · 2022-08-02
> **Related works and visualization experiment explain the scalar information loss**
>
> Thank you for your valuable comments. We used notations Q1 and Q2 for the questions in our response.
>
> **Q1**  As illustrated in line 243-244 of our paper, it is argued that the scalar information learns intra-class variance. We also confirmed similar properties as shown in Figure 4 of our paper, which visualize the results of the experiment on a toy task with 2-layer neural network. For example, representations of the variants of the digit 8 in MNIST are located in the same direction with different scales in the Figure 4(b). If we use the only angular difference of the representations, all variants are regarded as a single representation projected onto a vector with a specific scale. Then, when we use the overlapped representation, we can not distinguish the difference between variants anymore. We guess that there other various information might be lost by projection beyond the intra-class variance, but it depends on the types of neural networks and applications. We tried to confirm the existence of such information loss and remove them regardless of what it is. To more directly show the information loss, we added detailed numerical results in terms of the loss in Table 9 (Appendix E).
>
> **Q2** We guess that this question is about the title of Table 2. If it is correct, we thank this suggestion and agree with the reviewer that the term "Robustness" causes confusion to "adversarial robustness," which is not our intention. To clarify it, we revised the term "Robustness on Network Width and Depth ..." to "Retention of Training Ability ... in Various Network Width and Depth Settings."

---

### Meta-Review · Area_Chair_NCn7 · 2022-08-27

**Recommendation:** Accept
**Confidence:** Certain

**Metareview:**

This paper proposes spherization layer by first transforming the pre-activations into angles, then transforming the angles into cartesian coordinates on a sphere, and finally training weight parameters without bias. The proposed spherization layer is geometrically meaningful, and is generally applicable. It is demonstrated on a range of experiments.

Reviewer v1RE, who gave a rating 5, pointed out two related references and asked for more explanation about motivation. The authors explained the difference between their paper and the two references, and explained the motivation. All other reviewers are positive about this paper.

**Award:**

No

---

### Decision · Program_Chairs · 2022-09-14

Accept